ISAnWin: inductive generalized zero-shot learning using deep CNN for malware detection across windows and android platforms

Tayyab Umm-e-Hani 1 ummehani.asif@gmail.com
Khan Faiza Babar 1
Khan Asifullah 2
Durad Muhammad Hanif 1
http://orcid.org/0000-0002-7023-7172 Khan Farrukh Aslam 3
Ali Aftab 4
1 Department of Computer & Information Sciences, Pakistan Institute of Engineering and Applied Sciences , Islamabad , Pakistan
2 Pattern Recognition Lab, Pakistan Institute of Engineering & Applied Sciences , Islamabad , Pakistan
3 Center of Excellence in Information Assurance, King Saud University , Riyadh , Saudi Arabia
4 School of Computing, Ulster University , Belfast , United Kingdom
Akleylek Sedat
Electronic publication date: 2024 Dec 23
Publication date: 2024
Volume: 10
Electronic Location ID: e2604
Received 2024 Sep 30; Accepted 2024 Nov 21
Copyright: © 2024 Tayyab et al.
Copyright year: 2024
Copyright holder: Tayyab et al.
License: This is an open access article distributed under the terms of the Creative Commons Attribution License, which permits unrestricted use, distribution, reproduction and adaptation in any medium and for any purpose provided that it is properly attributed. For attribution, the original author(s), title, publication source (PeerJ Computer Science) and either DOI or URL of the article must be cited.
License URL: https://creativecommons.org/licenses/by/4.0/

Keywords: Malware detection, PE malware, Android malware, End-point protection, Artificial intelligence, Deep learning, Zero-shot learning, Algorithms and analysis of algorithms, Data mining and machine learning, Data science

Funding: King Saud University, Riyadh, Saudi Arabia, under Researchers Supporting Project RSPD2024R1062 This research was funded by King Saud University, Riyadh, Saudi Arabia, under Researchers Supporting Project number RSPD2024R1062. The funders had no role in study design, data collection and analysis, decision to publish, or preparation of the manuscript.

==============================
Effective malware detection is critical to safeguarding digital ecosystems from evolving cyber threats. However, the scarcity of labeled training data, particularly for cross-family malware detection, poses a significant challenge. This research proposes a novel architecture ConvNet-6 to be used in Siamese Neural Networks for applying Zero-shot learning to address the issue of data scarcity. The proposed model for malware detection uses the ConvNet-6 architecture even with limited training samples. The proposed model is trained with just one labeled sample per sub-family. We conduct extensive experiments on a diverse dataset featuring Android and Portable Executables’ malware families. The model achieves high performance in terms of 82% accuracy on the test dataset, demonstrating its ability to generalize and effectively detect previously unseen malware variants. Furthermore, we examine the model’s transferability by testing it on a portable executable malware dataset, despite being trained solely on the Android dataset. Encouragingly, the performance remains consistent. The results of our research showcase the potential of deep convolutional neural network (CNN) in Siamese neural networks for the application of zero-shot learning to detect cross-family malware, even when dealing with minimal labeled training data.

Introduction

The increased usage of gadgets in every walk of life has exponentially uplifted the amount of data in the cyber world. This digital data is widely spread over multiple types of machines, from personal computers to data servers in massive cloud infrastructures, and mobile phones from a variety of vendors. Around the globe, researchers have been busy finding the solution to keeping data safe and secure, wherever it is stored. Starting from the misconception of Windows being the most targeted operating system by hackers, the world moved towards the belief that the Internet was the only source of malicious activity. With the passage of time and the exploration of researchers, all these misconceptions eventually died. A digital report by Statista (Taylor, 2023) states that the most commonly used operating system by desktop users is Windows, which is the major reason for it being a popular target of hackers. If we look into the statistics generated by StatCounter (2024) regarding the popular operating system across different devices, we conclude that Android is the most popular operating system followed by Windows, as shown in Fig. 1.

Figure 1 Popular operating systems across different digital devices worldwide.

The investigation carried out by cyber security specialists into the chaos created by malware led to various sophisticated types of malware other than viruses and different sources of malware injection instead of only the Internet. Commonly encountered types of malware include trojans, worms, adware, spyware, rootkits, ransomware, and keyloggers (TitanFile, 2024). The virus is a malicious piece of code that can get attached to any file/document with the evil intentions of data leakage or disruption. They can enter the system through any external drive without attaching the system to the Internet. In contrast, worms replicate themselves into a system on the provision of the network. Trojans are malware that disguise themselves as legitimate software to trick the victim into executing malicious software. Spyware seem to be a useful software capable of leaking data without the user’s permission, thus compromising his data privacy. Rootkits are capable of getting root-level access and can protect other malware by getting attached to them, thus hiding them from being detected. These rootkits can be introduced to a system via downloading any malicious email attachment or through any external drive (Tayyab et al., 2022).

With the continuous expansion of digital ecosystems, our dependency on interconnected devices has grown exponentially. Unfortunately, this surge in technological connectivity has also attracted malicious actors who deploy a vast array of sophisticated malware to compromise the security and privacy of end users (Ali & Khan, 2013). Traditional malware detection systems, while effective to some extent, often lack the adaptability to keep up with the rapidly evolving landscape of malware. Moreover, many existing solutions are limited to specific platforms, hindering their ability to provide comprehensive protection across different devices. End users despite having the edge of keeping their data across different devices are facing the threat of data leakage and data corruption from multiple platforms.

In our hyper-connected digital world, the threat of malware looms large, transcending the boundaries of operating systems and devices (Derhab et al., 2023). From desktop computers to smartphones, each platform presents a unique challenge in the battle against malicious software. Yet, the quest for a universal solution remains elusive, as the diversity of platforms, coupled with the dynamic nature of malware, complicates detection efforts. The necessity for a unified and efficient malware detection system, capable of safeguarding a variety of devices, cannot be overstated. Another desirable feature of an efficient malware detection system is to be able to detect unseen malware, unlike conventional malware detection systems. The crux of the issue lies in the disparity among operating systems and their respective architectures, exacerbating the challenge of deploying a standardized malware detection system across multiple platforms. The relentless evolution of malware strains the efficacy of traditional detection methods, leaving users vulnerable to novel attacks.

Various approaches have emerged in the realm of malware detection, each with its strengths and limitations (Belaoued et al., 2019). Signature-based detection identifies known malware by matching against predefined signatures. Heuristic analysis extends this capability by discerning suspicious behavior based on predefined rules. Machine learning and AI-powered algorithms promise adaptive detection capabilities by discerning patterns in vast datasets, while behavioral analysis scrutinizes software actions in real time for signs of malicious intent. However, these solutions are not without their flaws. Signature-based detection falters against polymorphic malware and zero-day exploits, rendering it ineffective in combating emerging threats. Heuristic analysis, while more versatile, generates false positives and negatives. Machine learning approaches demand extensive training data and are susceptible to adversarial attacks, undermining their reliability. Meanwhile, behavioral analysis, while promising, imposes significant computational overhead, impeding its practicality on resource-constrained devices.

To address the problems of the availability of labeled data for new and unseen malware classes and after the zero-day malware problem, we adapt the technique of zero-shot learning (ZSL) that learns the general and high-level features related to knowledge over a series of similar tasks and makes use of a few samples from the classes seen during training to extract task-specific information. For the testing scenario, it comes across some malware families that are unseen during the training phase. To fulfill our objective, we use the Siamese Network with our custom designed a convolutional neural network (CNN) based ConvNet-6. Moreover, our proposed solution is thoroughly tested for detecting malware across different operating systems irrespective of the differences in their architecture.

In a broader spectrum of malware detection, following are the research questions that we attempt to answer through our proposed framework ISAnWin using the novel ConvNet-6: Is there any possibility of the existence of any model that can be used for detecting the unseen malware from existing malware families that can cater to polymorphism?

Can a zero day malware that does not belong to any existing malware family be detected to enhance the security posture of endpoints?

Can there be a unified and efficient malware detection system for safeguarding multiple devices running different operating systems?

In this article, we propose ConvNet-6, a CNN based model to address the problems of detecting and classifying unseen malware samples. A unified model is developed to work for both Windows and Android systems. The main contributions of this article are as follows: We implement visualization and data augmentation techniques in a systematic manner, significantly enhancing the representational capacity of malware datasets, which improves accuracy and generalizability of classification models and enables them to identify nuanced patterns of malware behavior more effectively.

A customized task-specific loss function tailored for measuring the similarity between malware samples is developed.

An innovative architecture based on ConvNet-6, a novel CNN-based model that exploits the strengths of CNN layers and the Average Pooling layer is proposed.

We propose an extensively tested architecture that is trained to learn the maliciousness in samples irrespective of the operating system platform.

The rest of the article is structured as follows: “Background and Related Concepts” presents the Background of malware detection approaches. “Theoretical Framework” is about the preliminaries of the method followed in this research. “Related Work” is about the work related to malware detection and classification for Windows and Android systems using AI. “Methodology” describes our model. “Experiments and Results” discusses the experiments and results. The “Conclusion” section concludes the research work done for this article and mentions the future direction of research.

Background and related concepts

Existing malware detection systems use multiple approaches to combat the chaos that can be created in case malware becomes successful. These approaches can be categorized according to their strategies of working. Figure 2 shows the broad categories of malware detection.

Figure 2 Malware detection approaches.

Signature based malware detection

A signature is a piece of pattern unique to each sample. This unique pattern can be calculated from the features extracted statically or dynamically. This is the most commonly adapted approach by the antivirus for malware detection. Antivirus systems store signatures in their databases. These signatures can be generated based on byte sequences, assembly instructions, strings, n-grams, flow graphs, and DLLs. Signature-based malware detection is fast but not efficient since it is easily hindered by obfuscations (Aslan & Samet, 2020).

AI based malware detection

Successful application of AI in numerous fields such as image processing, biotechnology, and e-commerce provoked researchers to exploit its strengths on malware detection as well. Since AI has emerged as the combination of machine learning and neural networks, the computing era has seen substantial work in malware detection using both streams of AI (Djenna et al., 2023). Figure 3 presents an overview of different techniques related to machine learning and neural networks that were incorporated in various research works.

Figure 3 AI-based malware detection.

Machine learning for malware detection

Two different learning methodologies have been used under the hood of machine learning: supervised and unsupervised. Supervised machine learning-based method is composed of two phases: (1) training and (2) testing. For carrying out the training phase, a big labeled dataset is required from which features are extracted either without executing samples or after executing them in a controlled environment. This feature extraction process is followed by feature preprocessing and feature selection, which resultantly produces a feature vector ready to be input to the machine learning-based heuristic engine for classifying samples into malicious and benign ones. Finally, this trained model is used to detect maliciousness in unknown samples during the testing phase. Before inputting the unknown sample to a trained machine learning-based heuristic engine, a sample has to undergo the process of feature extraction, preprocessing, and feature vector generation.

In contrast to supervised learning methodology, unsupervised learning-based malware detection using machine learning makes use of unlabelled datasets and works over the data structure. Unsupervised machine learning algorithms group together the samples based on similarities in the data structure. Like supervised machine learning algorithms, they also need feature extraction, preprocessing, and selection before being applied to the dataset.

Neural networks for malware detection

In recent years, neural networks have gained a lot of importance due to their inherited capabilities of automated feature engineering. CNNs are famous for their representational capacity. They make use of their hidden layers to learn the features of given samples (Khan et al., 2023). CNN architecture comes in different taxonomical structures depending upon spatial exploitation, depth, multi-path, width, feature map exploitation, channel boosting, and attention networks (Khan et al., 2020). Following supervised learning methodology, CNN, deep neural networks (DNN), and recurrent neural networks (RNN) have been extensively used for malware detection. When it comes to unsupervised learning, Neural Networks have manifested themselves in the form of Autoencoders, generative adversarial networks (GANs), and deep belief networks (DBNs) for malware detection (Aslan & Samet, 2020).

All these approaches for malware detection come with some shortcomings. Widely applied approach of signature-based malware detection easily fails on encountering obfuscated and encrypted samples. AI-based detection approaches which have been extensively used so far have proved to learn better when trained with fairly a large dataset. In case of any new malware type, and rarely encountered samples, a small dataset can cause the AI-based models to severely overfit, and high false positive rates (FPRs) are experienced (Tang, Wang & Wang, 2020).

Theoretical framework

AI has been proven to be an ever-evolving field. This evolution of AI has helped researchers in the field of malware detection and classification as well. One of the recent trends and techniques that the world of research has come across for malware detection and classification includes few-shot learning, which comes with its different variants.

Few-shot learning

Few-shot learning is a machine learning paradigm that focuses on training models to recognize and classify new classes with very limited labeled data. In the context of malware classification, few-shot learning can be applied when dealing with new and unseen malware families or variants that were not part of the training dataset. This capability is crucial because malware authors are continually creating new variants to evade detection (Tayyab et al., 2022). Formally, the main objective of FSL is to learn a classifier “h” capable of predicting label yi for each input xi. To find an optimized “h” FSL is trained on Training Dataset Dtrain and tested on Test Dataset Dtest. Typically, this quest of finding a classifier is facilitated by a limited number of samples in Dtrain. Normally, Few-Shot classification is quoted as N-way K-shot classification where N represents the no. of classes used during training, and K signifies the no. of samples used for training per class. Hence Eq. (1) shows the total no. of samples in Dtrain.

(1) Dtrain=(K∗N)samples

When there is only one sample used per class during training, i.e., K = 1, then Few-Shot classification is named as One-Shot classification whereas if Dtrain does not contain any sample of classes which are found in Dtest then FSL becomes Zero-Shot Learning (ZSL) (Wang et al., 2020). Since our primary goal of the research was to develop a model for the detection of malicious samples belonging to the malware class that our model would never have come across during training, therefore, we utilized the strength of ZSL.

Zero-shot learning

ZSL refers to the phenomena of classifying objects of unseen classes in the target domain by utilizing the knowledge obtained from seen classes in the source domain (Pourpanah et al., 2022; Cao et al., 2023; Ren et al., 2023). This objective of bridging the gap between seen and unseen classes is achieved by ZSL through semantic information. ZSL techniques can be broadly categorized into conventional and generalized ZSL based upon the settings of Dtest. As shown in Fig. 4A conventional ZSL-based techniques test the trained model on Dtest which contains only unseen classes. Generalized zero-shot learning (GZSL) based techniques (Gowda, 2023; Zhang & Feng, 2023) are more practical as they deal with Dtest that contains objects from both seen and unseen classes as shown in Fig. 4B.

Figure 4 Categories of zero-shot learning.

Since ZSL is a concept built upon the principle that no information about unseen classes has to be used during the training phase, therefore, semantic information is used which is the major key to building relationships between seen and unseen classes. It must contain the recognition properties of unseen classes obtained from the seen classes. Semantic information builds up semantic space, which contains both seen and unseen classes and thus can be used to perform ZSL and GZSL (Pourpanah et al., 2022; Yang et al., 2023).

In most cases, GZSL strives to learn embedding/mapping functions to map low-level visual features of the seen classes with their semantic vectors. This learned function is ultimately used to identify the novel/unseen classes by measuring the similarity between the prototype and predicted representations of data samples in embedding spaces. It is expected that in semantic space, similar classes have similar semantic vectors, but in visual space, there can be huge variation for similar classes.

Embedding space for GZSL can be categorized into semantic, visual, and latent embeddings (Rao et al., 2023).

Semantic embedding

The objective of semantic embedding models is to learn the forward projection function from the visual space to the semantic space. In semantic embedding, classifications are done in semantic space. Figure 5A presents the schematic view of semantic embedding.

Figure 5 Categories of embedding space for GZSL.

Visual embedding

A reverse projection function to map semantic representations back to the visual space is learned by visual embedding models. Using visual embedding, classification is done in visual space and its goal is to make the semantic representation close to their corresponding visual features. Figure 5B shows the schematic view of visual embedding.

Latent embedding

In latent embedding, both visual and semantic embeddings are mapped onto a latent embedding space to explore some common semantic properties across both modalities. Figure 5C exhibits the schematic view of latent embedding.

If we categorize GZSL based on approaches adopted during the training phase then GZSL can be inductive learning-based or transductive learning-based.

Inductive learning-based GZSL refers to the traditional method of supervised learning for training where labeled information having visual features and corresponding semantic vectors of seen classes is used for training. Dtest contains the samples of unseen classes for testing the generalizability of the GZSL model over unseen classes.

In contrast to inductive learning, the transductive learning-based approach for training the GZSL model trains the model with labeled visual and semantic information of seen classes in addition to semantic information and unlabeled visual features of unseen classes as well (Sarhan et al., 2023).

Siamese neural network

The Siamese neural network (SNN) is an architecture comprising two or more identical subnetworks, each incorporating neural networks designed to learn the semantic similarity between input samples. These identical subnetworks are configured with the same parameters and weights, ensuring that any parameter updates during the training phase are uniformly mirrored across all subnetworks. SNN functions as a task-invariant embedding model and operates as a distance metric-based meta-learner. Each subnetwork’s neural network extracts features and generates embedding vectors, with a similarity function employed to compute the similarity between the embedding vectors from each subnetwork. Figure 6 illustrates the fundamental architecture of an SNN with two identical subnetworks. SNNs demonstrate robustness to class imbalance and are focused on learning the semantic similarity between pairs of samples, making them suitable for implementing inductive learning-based GZSL. Different loss functions can be used to train SNNs for calculating the similarity between inputs. The most commonly used loss function to train twin subnetworks is the Contrastive Loss function shown by Eq. (2).

(2) L=Y∗D2+(1−Y)∗max(margin−D,0)2

Figure 6 Basic architecture of Siamese twin neural network.

Related work

This section provides readers with insight into research work done for detecting and classifying malware in Windows and Android using AI. This section is divided into two subsections. The first subsection highlights the work done to detect and classify malware in Android using different AI techniques, whereas the other subsection is dedicated to malware detection and classification in Windows using different AI techniques.

AI-based approaches for detecting and classifying malware on android systems

Mahdavifar et al. (2020) took advantage of easily available unlabeled data and applied the Semi-Supervised Learning technique. They used dynamically extracted features. Evaluation of their proposed model was done on the CICMalDroid2020 dataset. They used 13,077 samples for training their model. Their experimental results showed the capability of their model to classify Android apps concerning the malware category with an F1-Score of 97.84 percent and an FPR of 2.76 percent. This research work considered dynamically extracted features only, which limits its ability to work in real-time environments, and was not tested for detecting unseen malware classes and unseen malware samples.

Mahdavifar et al. (2020) carried their research further and used the Pseudo-Label Stacked Autoencoder in Mahdavifar, Alhadidi & Ghorbani (2022) which was trained by labeled and unlabeled datasets as it was following a Semi-Supervised learning approach. Hybrid features were used by them to train their model. They used 17,341 samples that were collected from December 2017 to December 2018. They achieved 98.3% accuracy on test data. One of the limitations of their research was the usability of the model in real-time environments due to the process of feature extraction dynamically. Secondly, they trained their model with a large number of samples per class which means their model was not trained for unseen malware families or malware families with few samples.

Hadiprakoso, Kabetta & Buana (2020) performed hybrid analysis by using the static features of the malware genome dataset and the Drebin project. For dynamically extracted features, they used CICMalDroid2020. Feature vector consisted of 261 features. They used 165 benign apps from the Play Store and 146 malicious apps from VirusShare for testing their model. The Extreme Gradient Boosting (XGB) ensemble model proved to be the best and most efficient model according to their research. Their model is not applicable in the real-time environment due to the limitations of feature extraction dynamically and their model was not tested for unseen malicious apps.

MSFDroid is a lightweight multi-source fast Android malware detection model that was proposed by Peng et al. (2022). The authors used the approach of ensemble learning and proposed a soft voting method that is dynamically adjusted by the weights of each base model. Their architecture takes the Classes.dex and Android Manifest.xml files in the APK sample as input. The extracted features from these two files are used for prediction by four base models. Three publicly available datasets were used for training and testing. Extensive testing of the model in real-time devices with an accuracy of “97.2%” proved it to be a better fit for real-time environments. However, they did not test the usability of the model for the detection of unseen malware classes and samples.

Al-Fawa’reh et al. (2020) explored the strengths of visual representation of samples for better identification of malicious and benign patterns. They made use of CNN architecture to automate the feature engineering process and trained the model with a balanced dataset of 10,000 malicious as well as benign APKs. Their evaluation method did not replicate the practical scenario as malware detection systems have to face imbalanced data. Secondly, they did not target to detect unseen malware samples, which is one of the most important use cases of practical scenarios.

Tang, Wang & Wang (2020) proposed ConvProtonet to make use of FSL for addressing the problem of malware samples’ scarcity. They introduced an induction module for generating an appropriate class-level malware prototype for classification. They adopted a prototypical approach for applying FSL. In extreme training conditions, they used five samples per malware family to train the model. Furthermore, they tested their model’s generalizability by testing it over malware from datasets other than Android malware. Their model does not perform well when the number of classes increases to be tested.

DL-AMDet presented in Nasser, Hasan & Humaidi (2024) uses a hybrid deep learning model for Android malware detection. This architecture has been trained and tested on CICMaldroid20. It is composed of three modules. One of the modules that works over static analysis utilizes CNN for feature extraction from Android applications and BiLSTM for processing the extracted features for determining the long-term dependencies. Another module is utilized by DL-MDet for dynamic analysis using deep autoencoders for anomaly detection. Nasser, Hasan & Humaidi (2024) claim that DL-AMDet can provide immunity against zero-day attacks due to the capability of anomaly detection through a module based on dynamic analysis, but the claimed capability of detecting zero-day malware has not been tested on the unseen malware families.

Another research work presented by Zhu et al. (2023) came up with the image based architecture MADRF-CNN for end-to-end Android malware detection. It focuses on combating the anti-reverse engineering techniques and hindrances in malware detection due to code obfuscation. They did not work on detecting the zero-day malware.

Our proposed model ISAnWin is meticulously crafted to address specific shortcomings underscored in the review article written by Guerra-Manzanares (2023). One of the problems highlighted in the article is neglecting the hybrid feature sets’ power. Most of the ML based work done to detect Android malware by Ou & Xu (2022) utilizes static features since they are easy to extract and many of the datasets provide static features only. Research work that is dependent on static features is prone to be deceived by obfuscation techniques. Despite being time consuming and technically challenging, some of the ML based research work done by Muzaffar et al. (2022) has focused on dynamic features. Although dynamic feature based detection models can overcome the problems posed by obfuscation techniques but these models are dependent upon the user intervention during feature extraction through sandboxing. Our proposed model ISAnWin leverages deep CNNs, strategically avoiding the issues encountered by both static and dynamic feature extraction methods, including challenges posed by obfuscation techniques and human intervention.

The literature review explicitly shows the use of datasets collected from a single source in terms of devices for Android malware detection. Liu et al. (2020) collected dynamic features by executing the Android apps on the Android emulator. The model was trained and tested on the same dataset. Thus, results of such works challenge the validity of consistent behavior of trained models across different devices. We responded to this problem by testing our trained model across multiple platforms.

Our research work produces quite promising results in comparison to existing literature regarding the detection of maliciousness in APKs. We utilize the visual representations of APKs rather than focusing on specific features for training the model as done by Kong et al. (2022). Our CNN architecture captures the global features, which helps the model learn all those features that could add maliciousness even being not correlated with the features found in training samples. In comparison to the work of Kong et al. (2022) our model is trained with only one sample per sub-family of training classes and performs better. During the evaluation, our model exhibits constant progress over increasing the families in support and query set, unlike the results shown by Tang, Wang & Wang (2020). Moreover, our model is aggressively tested for detecting unseen malware.

AI-based approaches for detecting and classifying malware on Windows systems

Hsiao et al. (2019) experimented with Siamese Network to perform an N-way One-shot task for malware classification. They focused on the visual patterns by converting the malicious and benign PE samples to greyscale images. Their model performed better for small values of N during training and testing, but with the increase in the value of N, its performance started degrading.

Research work presented by Deng et al. (2020) proposed an attention-based transductive learning approach for classifying malware from unseen classes. They tested their attention-based transductive learning network on a malware database of 11,236 samples with 30 different malware families. The transductive learning approach for FSL does not represent a practical scenario due to which results of this model are not reliable with respect to its application in real-time environments.

Khan et al. (2023) proposed a relational network based on the one-shot learning model to address the issue of scarcity of available malicious samples from each class of PE malware. Their model was trained with five samples per malware family in one of the experiments and in another experiment, they trained their model with one sample per class. Their approach has paved the way for research applying the concept of ZSL for detecting the malware family completely unseen in the training phase.

Our work in this article attempts to address all the issues that were not dealt with in these previous research works. This article makes use of inductive learning based GZSL, unlike the work done by Deng et al. (2020), to train the model in an environment that represents the real-time scenario. This research work proposes the model for unseen malware classes, therefore, it addresses the issue of scarcity or unavailability of labeled samples of malware families as well. Since it is tested on malware families encountered on different platforms, its generalizability has been tested to be used for a unified solution of detecting malware across different platforms as well. Our proposed model shows constant performance using SNN even when the value of N increases in the testing scenario, unlike the results presented by Hsiao et al. (2019) which came up with a decrease in evaluation accuracy on increasing the value of N.

Methodology

ISAnWin proposed in this article is an inductive learning based GZSL model that utilizes the advantages of SNN. The major components of ISAnwin as shown in Fig. 7 are described in the following subsections:

Figure 7 Components of ISAnWin.

APK parser

APK files are quite similar to Java Archive (JAR) files. They are compressed packages and can contain multiple directories such as assets, res, lib, and META-INF, along with many more as per the need. These compressed packages of APK contain various files including AndroidManifest.xml, classes.dex, and resources.arsc. Out of all these files, compiled bytecode is available in classes.dex file only. Our parser module shown in Fig. 7A unzips the APK and extracts the classes.dex file.

APK to image converter

To exploit the benefits of extracting malicious and benign patterns through spatial information and hierarchical features, we converted the output of DEX parser into greyscale images as shown in Fig. 7B. We followed the process described in Algorithm 1 for converting the APK files to images. Images produced by this converter were of variable sizes depending upon the size of classes.dex. After a thorough analysis of the converted images’ sizes, we came to the conclusion that the biggest size of the available image was 300 * 300, therefore we resized all the images to square images of size 300 * 300 to avoid the loss of visual pattern. Resizing was done through transforms.resize function of Pytorch with bicubic interpolation. We chose square images for implementing batch integration. To avoid any loss of information, we preferred the ceil operation of math during the image conversion process.

Sub-family categorizer

The dataset used in this research work was curated till the malware family level. It had four malicious families and one benign family. By looking at the images in all families, various images with nearly the same visual pattern could be spotted and some were identified with quite variable patterns within the same family. This variance in the visual pattern of images within the same family can be due to sub-families within the same malware family. In order to divide the families available in this dataset into subfamilies, we used the Average Hash (aHash) algorithm. The aHash is a non-cryptographic hash algorithm that converts the image into a short alphanumeric string. Unlike cryptographic hash algorithms, this hashing technique can identify even the slightest variation in the visual pattern of images. Visualization of samples after categorizing images within the same family into multiple subfamilies can be seen in Fig. 8.

Figure 8 Subfamily categorization.

Embedding module

The embedding module designed in ISAnWin as shown in Fig. 7D is an SNN that has two identical subnetworks with ConvNet-6 for producing the embeddings of input. ConvNet-6 used in subnetworks had six convolutional layers. The first 5 layers are followed by average pooling and ReLU. Input to each convolutional layer goes through the convolutional process shown in Eq. (3) for the calculation of the kth feature map.

(3) yk=wk∗x

where x denotes the input to the convolutional layer, yk is the kth output feature map and wk represents the convolutional filter associated with kth feature map. In our model, wk is 7 × 7 for 1st convolutional layer, then 2nd and 3rd convolutional layers use the convolutional filter of 4 × 4 whereas at 4th and 5th convolutional layers, convolution is done through the filter of size 3 × 3 and finally last convolutional layer has used the filter of size 2 × 2. ‘*’ shows the 2D convolutional operator. The kth output feature map of Eq. (3) goes to the average pooling layer. Average pooling layers were used for dimensionality reduction and to give weightage to every type of feature in the feature map by catering translational invariance as well. As shown in Eq. (4), the average pooling layer takes the arithmetic mean of all the elements in the pooling region which in our model ConvNet-6 corresponds to 2 × 2.

(4) ykij=1|Rij|∑(p,q)∈Rijxkpq

where ykij is the output of the pooling layer applied on the kth feature map and xkpq is the element at location (p,q) in pooling region denoted by Rij. Pooling region Rij represents the local neighborhood around the position (i, j). — Rij— denotes the size of the pooling region. ReLU was used to deal with the vanishing gradient problem. It is defined by Eq. (5).

(5) f(x)=max(0,x)

Since in ConvNet-6, output of average pooling layer undergoes the non-linear transformation through ReLU, therefore we can rewrite Eq. (5) as follows:

(6) f(ykij)=max(0,ykij)

Substituting Eqs. (5) in (6) gives us the output after applying ReLU in ConvNet-6.

(7) f(ykij)=max(0,1|Rij|∑(p,q)∈Rijxkpq)

The combination of convolutional layers followed by average pooling layers and transformed by ReLU are then followed by two fully connected layers with a dropout of “0.5” in between. The output of both the twin networks is given as input to the similarity score generator. Figure 9 shows the output shapes of ConvNet-6 and Fig. 10 shows the diagrammatic representation of the ConvNet-6 that we designed.

Figure 9 Output of ConvNet-6.

Figure 10 Visualization of ConvNet-6.

Similarity score generator

The similarity score generator operates over the absolute difference between the outputs of twin networks. This similarity score was optimized by using the contrastive loss function as represented by Eq. (2). To customize this loss function, we replaced Euclidean Distance “D” in Eq. (2) with an absolute difference as shown in Eq. (8).

(8) DAbsolute=∥X−Y∥

where X and Y are outputs of ConvNet-6 based SNN. Derivation of our customized loss function can be done by substituting Eqs. (8) in (2) as follows:

(9) L=Y⋅D2+(1−Y)⋅max(margin−D,0)2

Replacing D with |X−Y|:

(10) L=Y⋅(|X−Y|)2+(1−Y)⋅max(margin−|X−Y|,0)2

In the context of the contrastive loss function which has been extensively proven to work well with SNN, we chose to replace Euclidean distance with the absolute difference to handle minimal differential values since samples of sub-families within a family exhibit small differential values. Our customized loss function is less sensitive to small differences, which provides a more straightforward and stable measure of the distance. The use of Euclidean distance can be disproportionately affected by relatively small differences when the squared terms become too minimal, which can lead to numerical instability or less precise gradient updates in the learning process. In contrast, the absolute difference used in our customized loss function avoids this issue. Due to the constant gradient of absolute difference, our customized loss function leads to faster convergence in the presence of very small difference values. For small values of difference where the two samples belong to different sub-families of the same family, our customized loss function offers more stable, interpretable, and computationally manageable measures. This resulting robustness is particularly useful in GZSL scenarios where learning has to be done from limited training data and small variations between embeddings.

Experiments and results

This section provides details of experiments and their experimental setup in the first subsection. The second subsection presents the results of experiments and compare the results with the existing models tested on the CICMalDroid and PE Malware for distinguishing between benign and malicious samples.

Experimental Details

To conduct the proof of concept, we reviewed three previous studies. The first study (Mahdavifar, Alhadidi & Ghorbani, 2022) proposed a pseudo-label stacked autoencoder to distinguish between malicious and benign APKs. Their research work was focused on the development of a model that gets trained with labeled and non-labeled data samples. They tested their model by dividing the data into 70-30 and 60-40 train-test splits and claimed to achieve a high value of accuracy. The second study presented in Kong et al. (2022) used Siamese network with CNN for detecting malicious APKs. Their source of the dataset and the collection source of CICMalDroid20 (Mahdavifar, Alhadidi & Ghorbani, 2022) are the same. They divided APKs into two groups: benign and malicious. Out of these two groups, they focused on a specific feature set of benign and malicious APKs. Those feature sets were given as input to the Siamese Network and the purpose of using the Siamese Network was to find the similarity and dissimilarity between two images. Another study that we reviewed was Hsiao et al. (2019) in which Siamese network with CNN was tested on different values of N-way for detecting malicious PE files. They applied Siamese Network with ConvProto-4 architecture.

Our proposed architecture ISAnWin used the novel ConvNet-6 architecture. Our novel ConvNet-6 architecture shown in Fig. 10 is composed of six convolutional layers and utilizes the strengths of avg. pooling layer with a dropout of “0.5”.

To visualize the positive implications of the proposed model, we conducted two experiments. The Intra-Dataset Generalizability Experiment was carried out to practically observe the applicability of the proposed model for detecting unseen malware samples. On the other hand, Inter-Dataset Generalizability Experiment was performed to check the generalizability of the proposed model for detecting malware across different platforms.

Datasets description

The proposed model was trained using the CICMalDroid20 dataset (Mahdavifar et al., 2020). This dataset is composed of five distinct families including Adware, Banking malware, SMS malware, Riskware, and Benign. It is a collection of 17, 341 APK files from several sources including VirusTotal, Contagio security blog, AMD, MalDozer, and other datasets.

Another dataset that was solely used in the second experiment was collected from Figshare. It is a data bank of malicious PE samples classified into 12 malware families. Malware files in this dataset were primarily collected from three main sources: Malshare (https://www.malshare.com/), Virusign (https://www.virusign.com/) and Dasmlaware (https://dasmalwerk.eu/). Summary of the collected dataset is shown in Table 1.

Table 1 Summary of dataset collected from Figshare.

Family	Total no. of samples	
Emotet	907	
Fareit	1,584	
Flystudio	363	
Gafgyt	1,485	
Gandcrab	390	
Icedid	459	
Lamer	692	
Mepaow	820	
Mirai	1,440	
Ramnit	337	
Razy	668	

Experimental setup

The experimental environment was built with Intel(R) Core(TM) i7-8750H CPU @ 2.20 GHz, Omen laptop 15. It had 16 GB RAM installed in it along with NVIDIA GeForce GTX 1060 6 GB GPU. The proposed model was implemented using the Pytorch library of Python 3.9 in Anaconda Navigator.

Intra-dataset generalizability experiment

Our first experiment was conducted to visualize the performance of ISAnWin in detecting unseen malware. In this experiment, we performed meta-training and meta-testing on the same dataset of CICMalDroid20. Out of four malicious families, samples from two malicious families were used for training along with the benign samples. This experiment was repeated for 200 epochs with a learning rate of “0.0005”. The training and validation batch size was kept at four whereas testing was done on batch size of 1. Table 2 details the parameters during training of ISAnWin.

Table 2 Training parameters of ISAnWin and their values.

Parameters	Values	
Epochs	200	
Learning rate	0.0005	
Training batch size	4	
Validation batch size	4	
Trainable parameters	69,952,066	

After grouping the images in each category as a subfamily through aHash, we selected one image per subfamily for training purposes. We repeated this experiment with three different architectures in the Siamese network. One of the architectures, ConvNet-4 was based on CNN with four convolutional layers followed by max. pooling layer. A contrastive loss function was used to optimize the training of ConvNet-4 based SNN. Another concept was implemented with the same hyperparameters to check the applicability of transfer learning by using the pre-trained ResNet 34. It performed better than the ConvNet-4 based SNN but took considerable time for training. A novel architecture, ConvNet-6 was developed in our study to categorically focus on the features that distinguish a malware sample from another malware sample belonging to a different class. This novel architecture was designed as a deep CNN architecture where we used six convolutional layers and each layer was followed by an avg. pooling layer. Figure 9 shows the summary of our designed CNN architecture ConvNet-6 which performed quite well. This summary was produced using the Keras. Our proposed novel ConvNet-6 proved to be a promising architecture for the differentiation between benign and malicious samples. Use of avg. pooling layer after each convolutional layer drastically helped in focusing every kind of feature irrespective of the spatial density of features. Moreover, we customized the design of the loss function where the contrastive loss was used with absolute distance due to a very small difference between the samples of the same malware family. Our design of loss function proved to be successful as ConvNet-6 based SNN produced promising results when tested on seen and unseen samples of seen and unseen malware families. Figure 11 shows the comparison of ConvNet-6’s performance with different experimented CNN based architectures.

Figure 11 Comparison of Siamese neural network’s performance with different experimented CNN-based architectures.

Since we used the Inductive settings of generalized zero-shot learning, therefore, our Siamese Network trained with the ConvNet-6 shown in Fig. 10 was tested on five families, out of which two malicious families were kept unseen during training. For the families that were used during training, we tested the model on the samples that were not exposed during the training phase.

Inter-dataset generalizability

We performed a second experiment to test the generalizability of our proposed model ISAnWin with novel ConvNet-6 based SNN on malware datasets of different platforms. Therefore, we tested the model trained in an intra-dataset generalizability experiment with ConvNet-6 on the PE malware dataset. We tested the experiment on different malicious PE families and our model produced state-of-the-art results.

Results and discussion

Our proposed model ISAnWin was trained on three out of five malware families of CICDroidMal20 and was tested on all five malware families of the same dataset in Experiment 1. In Experiment 1 ISAnWin was tested on samples of all the malware families of CICDroidMal20 which depicts the real-life scenerio where seen and unseen malware samples both are encountered. The testing scenario classifies our proposed model as a zero-shot learning based model since it was tested on unseen malware families and unseen samples of seen malware families. We used multiple performance metrics to evaluate the performance of our proposed model during the training and testing phases. These performance metrics include accuracy, false positive, true positive, false negative, true negative, precision, recall and F1 score. The accuracy achieved by our proposed novel architecture depicts that the systematic use of average pooling layers after every convolutional layer efficiently learns the global and local features. Moreover, we used customised loss function that utilized absolute difference rather than Euclidean distance in contrastive loss since the difference between the malicious samples was too small. This customized loss function that was used to train the ISAnWin effectively optimized the proposed architecture.

We conducted Experiment 1 with our novel ConvNet-6 based SNN using various threshold values, 0, 0.05, 0.1, 0.15, 0.2, 0.25, 0.3 till 1.0 to determine the optimal threshold for achieving improved results. Figure 12 shows the performance metrics achieved over different threshold values that were experimented with.

Figure 12 Performance achieved over multiple threshold values for experiment 1.

The confusion matrix for Experiment 1 with our novel ConvNet-6 based SNN over various threshold values is shown in Fig. 13.

Figure 13 Confusion matrix of ISAnWin with novel ConvNet-6 over multiple threshold values.

ROC for the novel architecture ConvNet-6 used in ISAnWin is shown in Fig. 14. The area under the curve (AUC) for the ROC curve is approximately “0.695”. After thorough experimentation performed for the selection of optimal threshold value, we concluded that the model trained on “0.3” as the threshold performs better.

Figure 14 ROC of ISAnWin with novel architecture ConvNet-6.

Results of Experiment 1 implemented with different architectures in SNN are shown in Table 3 with the best accuracy value shown in bold. Figure 15 shows the training and loss validation graph of ConvNet-6 architecture used in the Siamese Network.

Table 3 Results of intra-dataset generalizability experiment.

The bold entry shows the best accuracy value.

Architecture	Threshold	Accuracy	Precision	Recall	F1	
ConvProto-4	1.0	0.73	0.39	0.65	0.49	
ConvProto-4	0.8	0.76	0.42	0.54	0.47	
ConvProto-4	0.3	0.77	0.42	0.54	0.47	
ConvNet-6	0.3	0.82	0.56	0.54	0.55	
ResNet-34	0.3	0.81	0.54	0.41	0.47	

Figure 15 Training and validation loss graph for novel ConvNet-6.

For the inter-dataset generalizability experiment, we tested the ISAnWin trained with ConvNet-6 implemented in SNN with “0.3” threshold. For testing the generalizability of ISAnWin on datasets from different sources and of different origins, we tried it on different values of N-way of PE malware. Our proposed novel ISAnWin was trained on CICDroidMal20 which consists of malware families found in Android whereas in Experiment 2 it was tested on PE malware. Existing research work of Hsiao et al. (2019) developed a SNN-based model with ConvNet-4 for detecting the malicious samples of PE but their model showed a decreasing value of accuracy on increasing the value of N during testing.

Comparison with PLSAE, FCSCNN and Siamese CNN

We carried out the comparison of our model’s performance from different perspectives with different models. The latest work done using the CICDroidMal20 dataset is presented in Mahdavifar, Alhadidi & Ghorbani (2022) which has used a pseudo-label stacked auto-encoder (PLSAE) based model for detecting malicious APK. They have trained their model on all four malicious families and tested it on the same four families. They achieved an accuracy of “97%” for detecting the malicious sample if it belongs to any of the four classes for which it was trained. Whereas ISAnWin was trained on one sample per sub-family of two malicious families of CICMalDroid20 and was tested on unseen families of the same and different origins as well and achieved an accuracy of “83%” and “82%” respectively.

Another comparison was made with the research work in Kong et al. (2022). Authors of Kong et al. (2022) have used Siamese network with CNN for detecting malicious APKs. Their source of the dataset and the collection source of CICMalDroid20 (Mahdavifar, Alhadidi & Ghorbani, 2022) are the same. They divided APKs into two groups: benign and malicious. Out of these two groups, they focused on a specific feature set of benign and malicious APKs. Those feature sets were given as input to the Siamese network and the purpose of using the Siamese Network was to find the similarity and dissimilarity between two images. They did not utilize the strength of CNN for automated feature engineering and did not exploit the application of the Siamese network for detecting unseen malware by calculating the similarity between seen and unseen samples, whereas our proposed architecture ISAnWin was designed and aggressively experimented to detect zero-day malware. A comparison of results is shown in Table 4. Figure 16 shows that our proposed ISAnWin performed better than the work of Kong et al. (2022) after being trained on just one sample per sub-family of malware. Despite the usage of SNN in both the works, ISAnWin is not only performing better than FCSCNN but has outclassed (Kong et al., 2022) by producing the comparable results with less computational cost.

Table 4 Performance comparison of ConvNet-6 with FCSCNN for detecting malicious APKs.

	FCSCNN (Kong et al., 2022)	ConvNet-6	
Dataset	Google Playstore and Virusshare	CICMalDroid20	
Training samples	3,000	799	
Tested on unseen families	No	Yes	
Accuracy	58%	83%	

Figure 16 Comparison of ConvNet-6’s performance with FCSCNN w.r.t no. of training samples.

Research work presented in Catak et al. (2021) has made use of data augmentation to improve the performance of CNN based model for malware detection. They injected three different types of noise in samples of malware families which made the model learn the local features of all families strongly resulting in the high accuracy of malware detection for existing malware families. This overfitting is due to learning the local features whereas ISAnWin extracts local as well as global features in complex tasks to generalize on unseen samples of existing malware families as well as on unseen samples of unseen malware families.

Another comparison was made with the work presented in Hsiao et al. (2019) in which Siamese network with CNN was tested on different values of N-way for detecting malicious PE files. They applied Siamese network with ConvProto-4 architecture and their testing of the model on different values of N shows that their model did not perform well on increasing the N value in the support set, whereas our model showed consistent accuracy on increasing the value of N in the support set. A comparison of results is presented in Table 5. In comparison to Hsiao et al. (2019) our proposed architecture showed steady performance during the test phase, on increasing the value of N. For N = 1 its accuracy was “0.83”, for N = 2 accuracy was “0.82”, and for N = 3 to 5, it achieved a consistent accuracy of “0.81”. A comparison of ISAnWin’s performance with the Siamese CNN proposed by Hsiao et al. (2019) over the increasing value of N is shown in Fig. 17. Inductive learning based generalized zero shot training of ISAnWin which was optimized through a loss function customized to be used for the dataset containing malware, produced an effective and efficient model capable of capturing malicious features irrespective of the OS platform where it is being utilized for the detection of malware.

Table 5 Performance comparison of proposed ConvNet-6 in ISAnWin with Siamese CNN with for N = 5 in test dataset for detecting PE malware.

	No. of families in training phase	K Value	Accuracy	
Siamese CNN (Hsiao et al., 2019)	35	3	75.3%	
ISAnwin: ConvNet-6	3	0	81%	

Figure 17 Comparison of ConvNet-6’s performance with Siamese CNN’s performance over increasing values of “N”.

Figure 18 shows the comparison of ConvNet-6’s performance with FCSCNN and Siamese CNN. It is evident from Fig. 18 that ISAnWin being a Generalized Zero-Shot model is capable of detecting unseen malware found on different computing platforms. ISAnWin has performed better than Kong et al. (2022) and Hsiao et al. (2019) due to the ConvNet-6 which has proved to capture local as well as global features found in malicious samples. The customized loss function has helped ISAnWin in being optimized for calculating the difference between two input samples.

Figure 18 Performance of ISAnWin:ConvNet-6 over multiple datasets.

Comparison of ISAnWin using novel ConvNet-6 with existing CNN architectures used for few shot learning and its variants based architectures

Hsiao et al. (2019) introduced CNN architecture in SNN with four convolutional layers and max pooling layer following each convolutional layer. Their use of max pooling layer is primarily focusing on the dominant features which are the distinguishing features for the samples of different malware families. However, the difference of features in the samples of different sub-families within the same malware family exhibit small amount of difference or appear as outliers among the prominent features, and are thus pertinent to be paid attention to, which are not being captured by their proposed architecture. Our study has carefully catered to the small amount of variation between samples belonging to different sub-families within the same malware family. Moreover, the use of Manahattan distance in Hsiao et al. (2019) has overshadowed the geometrical distance between the two samples. Further testing of their trained model did not prove to work well on increasing the families in the test scenario as shown in Fig. 17 whereas our proposed ConvNet-6 in ISAnWin has performed well in the testing scenario on increasing the no. of malware families as can be seen in Table 6. Another research work presented in Cen et al. (2024) has used a fully connected autoencoder containing two fully connected layers in encoder, which was designed to focus the learning of core features of malware families without catering to the less dominant features that can be the distinguishing features between the samples of sub-families of same ransomware family. Dense layers treat all input features equally and do not preserve spatial relationships. This can lead to the loss of important spatial hierarchies and patterns in data, which are crucial in malware detection where the arrangement of features matters. Whereas, our proposed CNN based architecture: ConvNet-6 can capture local and global spatial hierarchies through convolutional and pooling layers, making it more effective for extracting meaningful features from data like images. In the case of GZSL where a limited amount of data has to be used for training purposes, dense layers result in overfitting due to a large number of parameters, and to cater to the issue of overfitting, we have made use of the average pooling layer after each convolutional layer to get useful distinguishing features extracted. Due to the lack of ability to exploit local patterns and contextual information, a two-layer AE might not generalize well to new data, especially in complex tasks where the trained model encounters an unseen malware belonging to an unseen malware family that has slight differences in features in comparison to the existing families. In contrast to the architecture proposed in Cen et al. (2024) our proposed architecture is capturing local patterns as well as contextual information. ConvNet-4 proposed in Tang, Wang & Wang (2020) focuses high level features only and produces feature vectors by choosing the maximum value of each feature map, thus ignoring the less dominant features or important features found in scarcity and represented as outliers. Our proposed CNN based architecture is designed to extract local as well as global features to learn all the relevant features required to distinguish between the samples of different malware families as well as different sub-families of the same malware family. ConvNet-4 has used cosine similarity for calculating the distance between the two samples. Since cosine similarity calculates the difference between two samples on the basis of the angle between their vectors, therefore, a little variation in the two vectors can be interpreted as a big angular displacement resultingly identifying the two samples of different malware families despite being part of two different sub-families of the same malware family. Keeping under consideration the level of variation among the samples of different sub-families within the same family and between the samples of two entirely different malware families, ISAnWin is designed to make use of absolute difference as a similarity score generator.

Table 6 Comparison of ConvNet-6 accuracy with Hsiao et al. (2019) across different testing scenarios over increasing no. of malware families in testing scenario.

CNN architecture	No. of malware families in testing scenario	
N = 1	N = 2	N = 3	N = 4	N = 5	
Siamese CNN	100	92.6	84	75	72	
Proposed ConvNet-6	83	82	81	81	81	

Conclusion

Malware has become the biggest source of data security breaches and with the widespread use of different computing platforms, malware is found in various shapes hidden in different types of files. It has become the need of the day to have a malware detection system capable of detecting unseen malware irrespective of the platform where malware has to be detected. Existing methods become insufficient either due to the scarcity of available data or due to specific feature-centric methods.

In this article, our proposed solution, ISAnWin using novel ConvNet-6, represented a significant advancement in the field of malware detection and zero-shot learning. Through the utilization of an inductive learning approach and SNN with our novel ConvNet-6 architecture, ISAnWin demonstrated its efficacy in both training on Android malware and testing on Android as well as PE malware. The inclusion of a novel ConvNet-6 design with average pooling layers enhanced the model’s ability to capture both local and global features, contributing to its robust performance. This research addressed several critical gaps in the existing literature by training the model with minimal malware samples, specifically targeting zero-day threats. Moreover, the model’s versatility is evident as it successfully detected malicious files across different operating systems, including Windows and Android.

Looking ahead, there are promising avenues for further research. Enhancing performance metrics such as precision and false negative rates (FNR) remains a priority, given the importance of accurately identifying malware. Additionally, future investigations could focus on extending the model’s capabilities to detect maliciousness in diverse file types, such as PDFs, with a particular emphasis on advanced persistent threats (APTs) (Mirza et al., 2014). APTs present unique challenges due to their distinct behavioral patterns, requiring tailored detection mechanisms.

Supplemental Information

Supplemental Information 1 One of the models.

Supplemental Information 2 Training Code.

Supplemental Information 3 Data Loader.

Supplemental Information 4 Division of Dataset into Training, Validation and Testing.

Supplemental Information 5 Customized Loss Function.

Supplemental Information 6 Testing Code.

Supplemental Information 7 APK to Image Converter.

Additional Information and Declarations

Competing Interests

Author Contributions

Data Availability

Farrukh Aslam Khan is the Associate Editor of the journal.

Umm-e-Hani Tayyab conceived and designed the experiments, performed the experiments, analyzed the data, performed the computation work, prepared figures and/or tables, authored or reviewed drafts of the article, and approved the final draft.

Faiza Babar Khan analyzed the data, prepared figures and/or tables, and approved the final draft.

Asifullah Khan conceived and designed the experiments, authored or reviewed drafts of the article, and approved the final draft.

Muhammad Hanif Durad conceived and designed the experiments, authored or reviewed drafts of the article, project Supervision, and approved the final draft.

Farrukh Aslam Khan analyzed the data, authored or reviewed drafts of the article, project Supervision, and approved the final draft.

Aftab Ali analyzed the data, authored or reviewed drafts of the article, project Supervision, and approved the final draft.

The following information was supplied regarding data availability:

The CICMalDroid2020 dataset (used for training and testing), PE malware data dataset used for testing, raw data and code for training the model are available at figshare: Tayyab, Umm-e-Hani (2024). ISAnWin. figshare. Dataset. https://doi.org/10.6084/m9.figshare.26323711.v2.

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
