# Peer review of "ISAnWin: inductive generalized zero-shot learning using deep CNN for malware detection across windows and android platforms"

_PeerJ Computer Science, doi:10.7717/peerj-cs.2604_

## Round 0.1 · original submission · Major Revisions

The reviewers have pointed out that important parts are missing. More details are still needed.

Reviewer 1 ·

Basic reporting

You presented a method for detecting unknown malware families using the Zero Shot learning method. However, I could not see a class belonging to zero-day attacks in the dataset. How did you detect zero-day attacks or how did you determine that they could be detected?

Experimental design

It is insufficient to compare your method with the ResNet-34 architecture. There are many CNN architectures. You should also compare it with architectures with different structures.

Validity of the findings

Has an ablation study been done? If you can obtain an accuracy and fscore, you can also add roc curves.

Reviewer 2 ·

Basic reporting

As pointed out below, the authors have not fully addressed the requested revisions.

- Why does the validation curve in the loss curve shown in Figure 13 end at the 100th epoch? This loss curve is not correct for proper training.

Experimental design

- AUC Metric also can be provided.

Validity of the findings

- Providing the confusion matrix would also help in better understanding the model's performance.

---

## Round 0.2 · accepted · Accept

Since the comments have been addressed, we are happy to inform you that your manuscript has been accepted for the publication.

Reviewer 1 ·

Basic reporting

The changes made are sufficient.

Experimental design

The changes made are sufficient.

Validity of the findings

The changes made are sufficient.

Reviewer 2 ·

Basic reporting

The revised manuscript can be accepted as presented.

Experimental design

The revised manuscript can be accepted as presented.

Validity of the findings

The revised manuscript can be accepted as presented.